# Gapless provides combined scaffolding, gap filling, and assembly correction with long reads

Stephan Schmeing[1,2] , Mark D Robinson[1,2]

Continuity, correctness, and completeness of genome assemblies are important for many biological projects. Long reads represent a major driver towards delivering high-quality genomes, but not everybody can achieve the necessary coverage for good long read-only assemblies. Therefore, improving existing assemblies with low-coverage long reads is a promising alternative. The improvements include correction, scaffolding, and gap filling. However, most tools perform only one of these tasks and the useful information of reads that supported the scaffolding is lost when running separate programs successively. Therefore, we propose a new tool for combined execution of all three tasks using PacBio or Oxford Nanopore reads. gapless is available at: https://github.com/schmeing/gapless.

## Introduction

High-quality assemblies are an essential part of many biological projects. For most well-studied organisms, high-quality references are available and large consortia work on improving them further, for example, the Telomere-to-Telomere (T2T) consortium (1 *Preprint*). Other projects, such as the Vertebrate Genomes Project (VGP) (2), work on expanding the availability of high-quality references, but often individual research groups need to create their own assemblies.

Long read technologies, for example, Pacific Bioscience (PacBio) SMRT and Oxford Nanopore sequencing, drastically increased the continuity, correctness, and completeness of many de novo assemblies. To include these reads in the assembly process, they can be directly assembled on their own (long-read only) or together with short reads (hybrid) or they can be used to improve a pre-existing assembly. Each of the three options has its own advantages and drawbacks. In this work, we focus on the improvement of preexisting assemblies and only briefly look at the other alternatives.

Long read-only assemblies need to address the higher error rates of long reads, with the exception of the new PacBio HiFi technology, which offers error rates comparable with Illumina short reads (3). A performant tool for such assemblies is **Flye** (4, 5). It applies an A-Bruijn graph to deal with the error rates and resolves repeats by identifying variations between the individual repeat copies.

Hybrid assemblies rely either on accurately placing short reads onto long reads for correction or on finding unique paths through the (short read-generated) De Bruijn graph using the long reads (6, 7). Furthermore, these assemblies require adjustments for two input types in case of adaptation to other sequencing technologies. For example, using PacBio HiFi reads as an accurate basis for the De Bruijn graph in combination with ultra-long nanopore reads for the graph traversal is not possible without changing the current tools. An example of this type of assembler is **MaSuRCA** (6, 8).

Scaffolding and gap filling of existing assemblies provide the largest flexibility, because any tool and technology can be used to create the initial assembly and the adjustment to the specific long-read technology can predominantly be outsourced to the mapper. However, the flexibility to handle any initial assembly prevents strong optimization on specific conditions that are possible if you control the complete hybrid assembly, such as having alternative haplotypes removed or repeated contig ends trimmed. Thus, the scaffolding strategy might perform worse than specialized assemblers that have control over the whole pipeline. **PBjelly** (9) was one of the first widely used scaffolding and gap-filling tools. It tried to assemble the reads overlapping a gap and was specifically designed for PacBio reads. State-of-the-art tools that can handle all types of continuous long reads (CLRs) include **LRScaf** (10) for scaffolding and **LR_Gapcloser** (11) and **TGS-GapCloser** (12) for gap filling. In particular, **LRScaf** bases its scaffolding on minimap2 (13) alignments. It only uses reads that map over the end of contigs/scaffolds in the original assembly and masks repetitive contigs, identified through high coverage. Connections are built from reads mapping to multiple contigs and only kept if the distance between contigs does not diverge strongly from the mean and if enough reads support the connection. From the remaining connections, **LRScaf** builds a graph and scaffolds along the connections, where the connected contigs do not have an alternative. In case of alternatives, it follows long reads through the graph to find unique connections between contigs on both sides of the complex region.

---

[1]Department of Molecular Life Sciences, University of Zurich, Zurich, Switzerland   [2]SIB Swiss Institute of Bioinformatics, University of Zurich, Zurich, Switzerland

Correspondence: mark.robinson@mls.uzh.ch

**LR_Gapcloser** fragments the long reads into chunks of 300 bases, called tags, and maps them with BWA-MEM (14 *Preprint*). After filtering tags based on consistency, a read is proposed for gap filling if it has at least a minimum number of consistent tags on one side of the gap. If it also has enough tags on the other side and the read distance matches the gap distance, it is used for gap closure. Otherwise, the gap is iteratively shortened from both ends with additional checks to detect overlapping reads from opposite sides. **TGS-GapCloser** aligns the long reads to the gaps with minimap2 and selects a maximum of 10 candidates for each gap based on a score built from the read's mapping identities and alignment lengths for the two flanking segments. It does not take the gap length into account, because many scaffolding techniques do not have the resolution to accurately predict the gap size. The candidates are error corrected either with short or long reads and realigned to the gaps. Finally, the corrected candidates with the highest scores are inserted into the gaps.

Interestingly, the state-of-the-art tools perform either scaffolding or gap filling, despite the strong connection of the two tasks in the case of long reads. Here, we propose a new tool that performs **gapless** scaffolding, meaning it uses the reads that identify a connection between contigs directly to close the gap. In addition, it breaks contigs if a large enough number of reads fail the filtering steps because of a divergence at the same position in the original assembly.

In parallel to this work, another tool called **SAMBA** was developed (15) that also combines all three tasks. It uses minimap2 to align the reads to the assembly and then filters on the alignment length, requiring at least 5 kb alignments and a maximum of 1 kb of unaligned sequences in case both contig and read continue. For every remaining connection between two contigs, a consensus is formed using **Flye**. Afterwards, contigs are combined on unique connections and paths that split and recombine, forming so-called bubbles. Bubbles are resolved iteratively by choosing the longest alternative. Finally, bridged repeats are handled by removing the repeated contigs and using the spanning reads to fill the gap.

## Results

### Overview of the gapless algorithm

**gapless** consists of four (Python) modules: split, scaffold, extend, and finish (Fig 1). The split module separates the scaffolds from the original assembly into contigs. The scaffold module is the main part of the program and performs the scaffolding, gap closure, and assembly correction. The optional extend module inserts consistent sequence from the long reads at the unconnected ends of contigs. If the reads offer two possible extensions for a contig end, they are added as separate contigs into the assembly. Finally, the finish module applies the list of changes from the scaffold and extend module to the split assembly and writes out the improved assembly as a FASTA file.

For convenience, a bash script is distributed with **gapless** to run the four modules in a pipeline. It requires two inputs: an existing assembly in FASTA format and long reads in FASTQ format. The

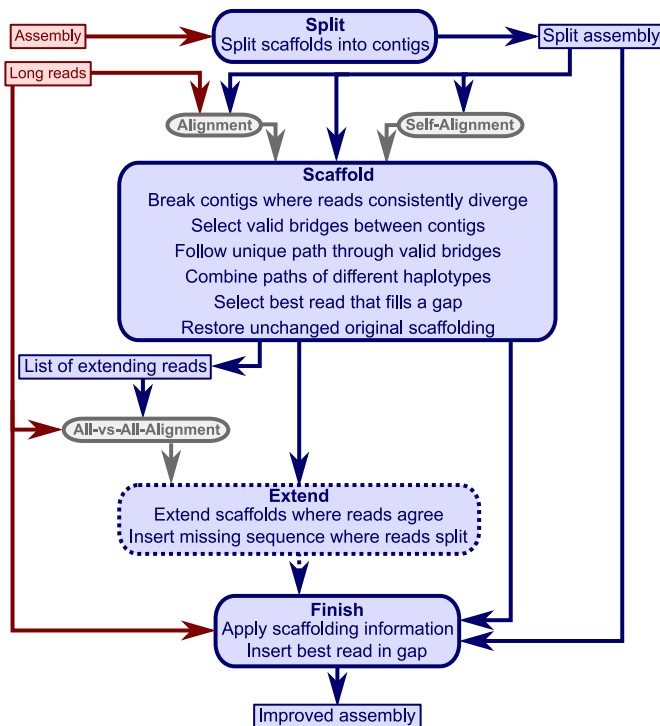

**Figure 1. gapless overview.**
The four modules of **gapless** (blue, rounded corners) and the mandatory inputs (solid, red). The extend module is optional. The alignments (grey) require a mapper such as minimap2. Two of the intermediate files and the final output file are included in the overview (blue, sharp corners).

pipeline applies minimap2 (13) to create three alignments: the split assembly to itself, the long reads to the split assembly, and a selection of extending reads to themselves. Seqtk (16) is used to filter the long reads and a final polishing step with racon (17) was added after the finish module. Furthermore, the pipeline supports multiple iterations to successively improve the assembly, and was constructed in bash to allow users to easily change the mapping software or parameters for better adjustment to changes in technology.

The scaffold module is the core of **gapless**. It requires the split assembly to extract the names and lengths of existing scaffolds, the alignment of the split assembly to itself to detect repeats, and the alignment of the long reads to the split assembly. The long read alignments are initially filtered, requiring a minimum mapping quality and alignment length, and in case of PacBio, only one Subread per fragment is kept to avoid giving large weight to short DNA fragments that are repeatedly sequenced multiple times. If the user provides CCS (circular consensus) reads, the Subread filter is not required. In a second step, mappings that are fully contained within a contig or stop aligning to the contig before the contig or read ends are removed. However, if multiple reads consistently diverge at the same position within a contig, the contig is split at this position instead of removing the mappings.

From the accepted alignments, bridges are formed. A bridge is defined by the two contig ends it connects and the read distance in between. Bridges with similar distances are clustered together. The

bridge clusters are filtered for chimeric connections, alignments to the wrong repeat copy, and rare connections potentially caused by sequencing errors. The remaining bridges are used for scaffolding. An example of the filtering is provided in Fig S1.

For four datasets with a coverage between 21× and 33×, the performance of the filters are plotted in Fig S2. More details can be found in Figs S3–S6. False discovery rate and Fraction Possibilities Kept are based on an alignment of the contigs to the respective reference assemblies. The bridges remaining after all filters up to the specified one are considered as positives. True positives additionally require that the orientation is correct and the distance is within 0.5 to 1.5 of its true value. The Fraction Possibilities Kept is the fraction of true positives existing before the filtering that is still covered after the filtering. Covered means in this case that the full new contig created by the combination of the two contigs is completely included in a new contig created from bridges surviving the filters. The combined filters clearly outperform a single filter on alignment length and the default values for the filters chosen in **gapless** perform well in all cases. However, for two datasets, even the most stringent filters do not achieve a false discovery rate below 23% (Nanopore) or 17% (PacBio). The conclusions drawn from these four datasets with a coverage between 21× and 33× still hold for datasets with very low or very high coverage (Figs S7 and S8).

After filtering, all contigs are scaffolded that do not have conflicting bridges. Then, we identify long-range connections from reads connecting at least three scaffolds and filter them in a similar manner to the bridges. From the remaining connections, a scaffold graph is built. After removing contigs that duplicate the sequence of another contig from the shared paths in the graph, we identify origin–extension pairs. Although the scaffold graph gives all valid paths starting at a given scaffold, the origin–extension pairs select the paths (extension) conditional on the scaffolds on the other side of the given scaffold (origin). We build an initial set of paths along unique origin–extensions pairs. Loops and inverted repeats are added by specialized functions to the set. Then, this initial set is iteratively merged and combined. The merging stacks paths with the same ending that likely represent alternative haplotypes and the combination connects the paths along the best overlaps. When this process converges, we remove completely duplicated paths and run the process again. When this process converges once again, we merge shorter paths into longer ones if we find a unique position and run the process one last time. Afterwards, we try to combine nonoverlapping paths, trim duplicated ends and circular paths, and remove unconnected contigs that were previously identified as a duplicate of another contig.

The final scaffold paths describe the (**gapless**) scaffolding of the contigs. Original scaffold connections between untouched contigs are kept as gapped scaffolds. To fill the gaps, we first map the reads to the scaffold paths and then select the best read for each gap to fill it.

## Comparison with state-of-the-art tools

We benchmark **gapless** against existing tools using various datasets from *E. coli*, dolphins, and humans (Table 1). The results are compared with the accompanying reference genomes using QUAST-LG (21). The *E. coli* and dolphin data are CLR from PacBio and the human data contain both PacBio HiFi and Oxford Nanopore. To test the influence of coverage, we subsampled the full datasets with seqtk (16) to four coverage levels, starting always from the full dataset and using different seeds. Because of its lower overall coverage, we created only three subsampled datasets of the PacBio HiFi data.

**gapless** is compared with **SAMBA**, **LR_Gapcloser,** and **TGS-GapCloser**. Because the latter two do not perform scaffolding, we applied **LRScaf** first, whereas **gapless** and **SAMBA** were run directly on the initial assemblies. For dolphin and human, we use an initial **supernova** (22) assembly that already contained many gaps before scaffolding with long reads. For *E. coli*, the initial assembly was created with **SH-assembly** (23), which does not perform scaffolding. The Nanopore data were tested, in addition to the **supernova** assembly, on a **Flye** assembly created from the PacBio HiFi data. To judge the results in a broader context, we also included **Flye** assemblies that are directly based on the (subsampled) data.

Fig 2 shows the achieved continuity measured in NGA50 versus the misassemblies reported by QUAST. For *E. coli*, the most noticeable observation is that the direct assemblies from **Flye** continuously improve with higher coverage, whereas the performance of **LR_Gapcloser** and **TGS-GapCloser** fluctuates in an apparently random manner. **SAMBA** does not seem to make significant changes to the initial assembly. **gapless** is the only gap-filling strategy that shows a general trend to increase continuity with increasing coverage and outperforms the direct competitors in correctness and continuity for most coverages. In contrast to *E. coli*, the dolphin gap-filling assemblies all show improved contig continuity with higher coverage, except for **SAMBA**. However, they also experienced a higher number of misassemblies. Thus, they still do not use the full potential of the higher coverage compared with **Flye**, which improves continuity and reduces misassemblies. From the gap-filling tools, **gapless** consistently achieves the highest contig continuity and still has fewer misassemblies than **TGS-GapCloser** for all coverage levels except very high. **LR_Gapcloser** does not appear to close many gaps, which might be caused by its check on the gap length. Most gaps in the scaffolded dolphin assembly are created by **supernova**, which does not have the accuracy of long reads to estimate the distance between contigs. **SAMBA** even decreases the contig continuity, which is likely because of contigs that are correctly separated at the misassembly, but are not rescaffolded. **gapless** seems to remove nearly all of the original scaffolds and thus has a lower scaffold continuity for low and very low coverage compared with the other gap-filling tools that do not attempt to correct the initial assembly and keep all scaffolds. In the human PacBio HiFi data, **Flye** achieves higher continuity and lower error rates already with 8× HiFi data and >7 times higher contig continuity with the full 33× coverage. Nevertheless, **gapless** outperforms the other gap fillers and achieves a >3-times higher continuity for the full coverage. For the human Nanopore data, the results are similar to the dolphin PacBio CLR data, when also starting from an initial **supernova** assembly. However, starting from the **Flye** assembly of the HiFi data, **gapless** introduces many errors with only moderate continuity improvements. In addition, **gapless** and **LR_Gapcloser** seem to

**Table 1. Benchmark data.**

| Species | Data | Coverage | Source |
|---------|------|----------|--------|
| *E. coli* | PacBio CLR | 113, 57, 28, 14, 7 | Public Health England reference collections (18) |
| dolphin | PacBio CLR | 86, 43, 21, 11, 5 | Vertebrate Genomes Project (19) |
| human | PacBio HiFi | 33, 16, 8, 4 | Telomere-to-telomere consortium (20) |
| human | Oxford Nanopore | 121, 61, 30, 15, 8 | Telomere-to-telomere consortium (20) |

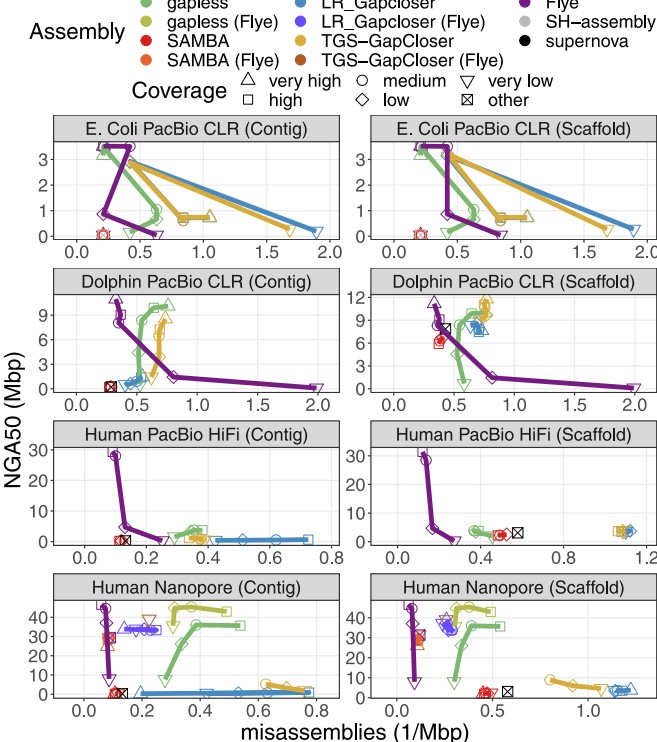

**Figure 2. NG50 versus misassemblies for the compared assemblies.**
The datasets are specified in Table 1. Human PacBio HiFi data do not have a very high coverage category. The "other" coverage category is the base assemblies improved with scaffolding and gap filling, which were created from a different dataset. Multiple assemblies were excluded because the methods crashed or did not finish within the one week limit of the cluster. Details are given in the Materials and Methods section.

predominantly increase the error rate and not the continuity with higher coverage.

Comparing the completeness and duplication ratio of the results (Fig 3), we notice that **gapless** has much lower duplication rates than the other gap-filling methods for all datasets except *E. coli*, where it is surprisingly the opposite. The duplications cannot only be caused by the initial assemblies, because they themselves have low duplication rates. This is especially evident in the case of the initial **Flye** assembly that does not have many gaps. Thus, the duplications cannot appear simply because of the gap filling, but must be caused to a large extent from the **LRScaf** scaffolding. Another observation is the strong drop of completeness for low and very low coverage assemblies with **gapless**. A manual check on the dolphin assembly with very low coverage showed that the racon polishing step causes decreased completeness by drastically

reducing the mapping rate of the contigs to the reference. For higher coverage, several percent higher genome completeness can be observed for the **gapless** scaffolding with Nanopore data compared with the respective **Flye** assembly. **LR_Gapcloser** and **TGS-GapCloser** counterintuitively lose completeness compared with the initial **supernova** assemblies for human data.

The relative time and memory requirements of the methods vary strongly between datasets (Figs S9 and S10). Generally, the direct assembly with **Flye** is time and memory efficient for low coverages, but performs worse at higher coverage than the gap-filling strategies. The HiFi dataset has a maximum coverage of only 33×, which partially explains that the initial **supernova** assembly already uses more time and memory than the high coverage assembly from **Flye**. The time requirements of **TGS-GapCloser** appear to scale poorly with coverage in the case of Nanopore data, because only medium coverage levels (and lower) finish within the one week limit of the cluster and only in the case of the initial **supernova** assembly. For the initial **Flye** assembly **TGS-GapCloser** finishes only in the very low coverage case. **gapless** and especially **TGS-GapCloser** have a strong increase in memory consumption for high coverage. The memory requirements of **gapless** are defined by the racon run at the end of each iteration and most time is either spent for the mapping or the consensus (Fig S11).

## Discussion

The strategy of the combined scaffolding and gap filling implemented by **gapless** successfully increases the contig continuity compared with other gap-filling tools and the additional correction step detects and removes misassemblies in the initial assembly.

**SAMBA** (15) does not work well in our benchmark. Therefore, we ran the 30× nanopore data also on the CHM13-WashU assembly from their article. Instead of the reported NGA50 of ~23 Mb, we only achieved ~9 Mb, which is the NGA50 of the input assembly. Thus, this result is consistent with what we observe in our benchmarks. We suspect that version 4.0.6 that we used has suffered in performance compared with version 4.0.5 used for their publication. This might be caused by the additional correction step that can split misassembled scaffolds introduced in version 4.0.6.

Direct assemblies with **Flye** outperform all gap-filling strategies in terms of correctness and continuity except for low and very low coverage. We believe that the lower performance is linked to repeats or haplotypes not represented as repeats in the original assembly. This assumption is consistent with the high rate of erroneous bridges even after the most stringent filters for the diploid dolphin assembly and the human HiFi assembly corrected with

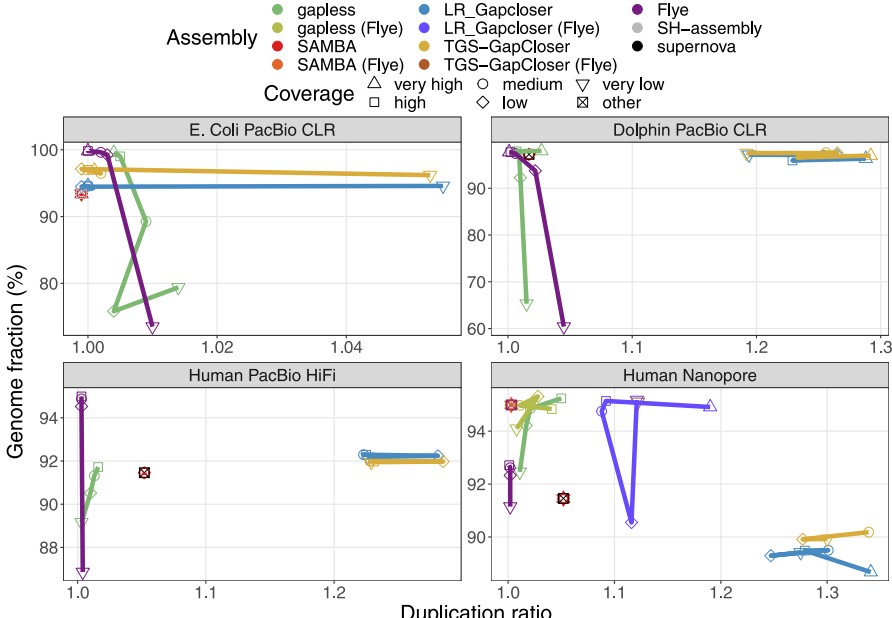

**Figure 3. Completeness versus duplication ratio for the compared assemblies.**
The datasets are specified in Table 1. We only show the results for the scaffolds. The results for the contigs are very similar. Human PacBio HiFi data do not have a very high coverage category. The "other" coverage category is the base assemblies improved with scaffolding and gap filling, which were created from a different dataset. Multiple assemblies were excluded because they crashed or did not finish within the one week limit of the cluster. Details are given in the Materials and Methods section.

nanopore reads, where only the difficult-to-resolve repeats remain as gaps. To achieve a similar performance as **Flye** for higher coverage, two strategies for further development of **gapless** are promising: (i) variant calling into the pipeline would allow to separate individual repeat copies as is done in **Flye** and to resolve haplotypes through trio binning (24); (ii) accepting mappings to the assembly graph would potentially improve the mapping in repeat regions and allow an improved handling of haplotypes over multiple rounds.

To reduce the time or memory requirements, little can be improved in the Python core of **gapless**, because the demands are largely determined by external software. A promising avenue to explore are changes to the polishing, either by not running it for every iteration, or by selecting the input reads and thus reducing the coverage racon has to deal with. For improvements with low coverage, it is also advised to use the data from the initial assembly for polishing or another tool, because the polished assembly from racon is purely based on the consensus of the reads and thus includes many errors if not enough reads are available.

Another extension would enable scaffolding with non-CLR, such as optical mapping from Bionano Genomics. This requires the functionality to turn off the gap filling and a special handling for contigs that are included in multiple scaffolds. It would allow low-coverage Bionano data to be used for scaffolding, which is not possible in the pipeline provided by Bionano Genomics, because it includes a de novo assembly step (25).

Overall, we presented a new tool for **gapless** scaffolding, that is, combined scaffolding and gap filling, that achieves better contig continuity than separate scaffolding and gap-filling tools. The included assembly correction can, in contrast to other scaffolding tools, remove errors in the initial assembly that are highlighted by the long reads. The necessary mapping and consensus calling are

performed with minimap2 and racon, but this can be quickly changed in the short accompanying bash script. **gapless** is especially useful for cases where direct long read assemblies are not applicable, either because the coverage is low (<15-fold) or the changes to the initial assembly need to be listed, for example, for changing the annotations.

# Materials and Methods

### Breaking contigs with consistently diverging reads

All mappings that stop aligning to a contig before the read or contig end are marked as potential breaks, and the position of divergence is stored. For every break, the number of breaks in close vicinity (including itself) is counted as support and all alignments that map continuously over the break region are counted as "vetoes." The support is compared with the vetoes according to the bridge selection rules in the next section. If the support is not removed by these filters, the break is accepted. The alignments of unaccepted breaks are removed, whereas accepted break positions in close vicinity are merged and the contigs are broken in the center of the merged break positions.

After breaking at those positions, all contig ends without alignments are trimmed, including the new ends from breaks. Only alignments of reads passing the initial filtering steps (mapping quality and alignment length) are counted for this. Contigs that are barely longer than the minimum alignment lengths after trimming are also removed to avoid on/off situations, where the accordance of a few bases decides if a read has an accepted alignment to the contig or not.

## Selecting bridges

From the accepted alignments, bridges are formed. A bridge is defined by the two contig ends it connects and the read distance in between. Bridges connecting the same ends are clustered based on the distance difference relative to the distance. The gap lengths from a potential scaffolding in the original assembly are not taken into account because they are often not very accurate. If multiple clusters connecting the same contig ends remain, they are treated the same as bridges connecting different ends. The few exceptions to this rule will be specifically mentioned. Bridges are also allowed from a contig to itself and even from a contig end to itself, but additional consistency checks are performed to avoid bridges caused by mapping artifacts. In the following, we will use the term bridge predominantly for bridge clusters with a count equal to the bridges in it.

Three filters are applied to the bridges. First, a minimum number of counts is required to remove bridges caused by chimeric reads. Then, the remaining read lengths on both sides are compared with remove alignments to the wrong repeat copy. Finally, **gapless** filters on the counts are compared with the expected counts for the mean distance of the bridge cluster. This removes rare connections potentially based on sequencing errors.

The remaining read length is almost identical to the alignment length because of previous filters on the mappings. The large exceptions are short contigs, where the reads exceed the contigs. To also handle these cases, we use the remaining read length. To test for alignments to the wrong repeat copy, we apply a one-sided Wilks-Rosenbaum exceedance test ([26]) that identifies bridges with a truncated length distribution on either side. In total, four tests per bridge are performed: the remaining read length is compared on each side of the bridge with other bridges that share this side and with other bridges that share the non-compared side. For each of the four tests, we identify the bridge with the longest remaining read length (control) and compare all other bridges (test) against this one. The Wilks-Rosenbaum exceedance test requires the number of reads belonging to the control and the test bridge and the count of control reads with a remaining length longer than the longest remaining length of the test bridge. When counting the latter, we adjust the test length with positive differences in bridge length. Bridges with a resulting $P$-value below a threshold in any of the tests are removed. In rare cases, all bridges of a contig end are removed. Thus, we restore bridges that connect two contig ends without any bridges starting with the removed bridge that has the largest minimum $P$-value (from the four tests).

To determine the expected number of reads for a given bridge distance, we partition the original, unbroken contigs into consecutive, nonoverlapping bins for various bin sizes. The bin sizes are chosen to represent the read length distribution with equal number of reads between them. The bins in a contig are shifted, such that all of them are full length, leaving an equal number of bases at each contig end uncovered. For each bin, we count the reads covering the full length and build one cumulative distribution of bins over counts per bin size (Figs S12 and S13). To exclude repeat regions, only the lower half of the cumulative distribution is fitted with the cumulative distribution function of the negative binomial distribution using a least square minimization. For every bridge, we

calculated the probability of the observed number of counts or less from the fitted cumulative distribution function with the lowest bin size still higher or equal to the mean bridge distance. Because of coverage fluctuations over the genome, we did not apply an absolute threshold, but instead, compare the bridges with all other bridges sharing a contig end. If a compared bridge has a many-times-higher probability, we remove the lowly probable bridge. To take the mapping quality into account, we performed this test on the cumulative counts going from high to low qualities. The highest quality that resolves a conflict is used. In case the two alignments forming a bridge have different mapping qualities, the bridges are first compared on the lower of the two qualities and only if those are equal on the higher ones.

## Building scaffold graph

After scaffolding contigs that do not have conflicting bridges, reads are selected that include at least three scaffolds connected by valid bridges. From these long-range connections, all consecutive occurrences of $n$ scaffolds are extracted with $n$ starting at 3. These long-range bridges are filtered based on the mappings to the outer scaffolds in the same way as the bridges between contigs previously with three exceptions: (1) the long-range bridges from an outer scaffold are only compared with other bridges sharing the remaining $n - 1$ scaffolds of the long-range connection; (2) the requirement of a minimum number of counts to avoid chimeric connections is explicitly lifted for this comparison, because long-range bridges are much less frequent than contig bridges; (3) both sides are required to not pass the filters to remove the long-range bridge instead of only one side. If only one side would be removed, we still have the shorter bridges connecting the outer scaffolds with the complex region in between and only lose information, but if we can remove both sides, an unlikely traversal of the complex region is erased. Afterwards, long-range connections are split at the outer scaffolds of a removed long-range bridge and the filtering is continued for long-range bridges covering $n = n + 1$ scaffolds.

The purpose of the scaffold graph is to list all supported paths for every scaffold. To build it, the split long-range connections are combined with the accepted bridges between scaffolds. For every scaffold in every connection, the paths up to the end of the connection in both directions are added as separate supported paths (Fig S14). The resulting graph is deduplicated and paths that are completely contained within longer paths starting at the same scaffold are removed.

## Removing duplicated contigs from scaffold graph

If two contigs have very similar sequences, we refer to them as duplicated. We identify them with the provided mapping of the split assembly to itself. Contigs are marked if they are covered to a large extent by an alignment from another contig that ends within a maximum distance from both contig ends.

If the sequences do not diverge enough between the duplicated contigs for the mapper to place the long reads unambiguously, all incoming and outgoing paths in the scaffold graph can go to both contigs. This undermines haplotype phasing and, if multiple duplications are in close vicinity, the possible combinations quickly

exceed the ploidy-based tolerance and prevent further scaffolding. Thus, we removed the duplicated contig with the lowest bridge counts from a shared path in the scaffold graph. If multiple contigs are duplications of different parts of another contig and the order and orientation match, they are also removed from the shared paths in the graph.

### Finding origin–extension pairs

Although the scaffold graph gives all valid paths starting at a given scaffold, the origin–extension pairs select the paths (extension) conditional on the scaffolds on the other side of the given scaffold (origin). Every pair of paths in the scaffold graph that have the same starting scaffold on different strands is a potential origin–extension pair. Every pair is considered twice, once with each of the paths as an origin. For filtering, we find the first diverging scaffold in a pairwise manner between origins of the same starting (center) scaffold and strand. From each of these branch points, starting at the furthest from the center, we follow all paths in the scaffold graph that are consistent with the origin until the center. All pairings with extensions that do not match the remainder of one of the consistent paths are removed. Fig S14 gives an example of a scaffold graph and all valid pairs. If a pair was filtered out, but the reverse (switched origin/extension) is valid, we also call it valid. This step and only checking at branch points reduces the effect of missing long-range connections, where only one of the alternative paths is found in the scaffold graph. This happens, when the reads are too short to reach the branch point, but the reads on the alternative connection do reach it.

It is important that the origin–extension pairs are consistent. Thus, we require that when we take a center scaffold and follow one of its extensions by one scaffold, the origins of the extended scaffold that go through the center scaffold have a valid pairing with the followed extension at the center scaffold. If these pairs are not present, we add them.

### Traversing the scaffold graph

Once we have the scaffold graph and origin–extension pairs prepared, we can search for the best paths to combine the scaffolds further. Fig 4 shows an overview and an example for this procedure. We start by finding non-branching paths that have not been scaffolded before, because parts of them are either repeated or present in multiple haplotypes. Loops and inverted repeats require each a specific handling, but all other paths are created by following unique extensions for a given origin. After the initial paths have been created, we add all nonunique origin–extension pairs as two-scaffold paths that contain the center and the first extending scaffold, except if they are already included in the paths from the loop or inverted-repeat handling. Completely unconnected scaffolds are added as single-scaffold paths.

The initial paths are merged and combined in the main loop. The first step merges paths that share both ends as long as the resulting paths do not have more alternatives then the specified ploidy. The second step combines overlapping paths if the overlap is unique or preferred over the overlap with other paths. These two steps are repeated until convergence is reached, when no paths can be merged or combined anymore.

After convergence is reached for the first time, all paths that are fully represented in another path are removed and the main loop is restarted. The second time convergence is reached, we tried to include all paths into another path based on the scaffolds at each end. If only one possible place is found for the insertion, we add the paths as an additional haplotype to the other paths. Afterwards, the main loop is restarted. The third and final time we reach convergence, the finishing procedure is started.

First, we extend all paths as far as they have a unique extension and try to combine paths again. We keep combined paths, but remove all extensions that did not allow a combination with another path. Afterwards, duplicated path ends and circular paths are trimmed and unconnected contigs are removed if they were identified as a duplication of another contig earlier. The final scaffold paths describe the (**gapless**) scaffolding of the contigs. Original scaffold connections between untouched contigs are kept as gapped scaffolds.

### Creating paths following unique origin–extension pairs

We group all origin–extension pairs from the same center scaffold that share an extension or an origin. To be considered for the path creation, a group must share the first extension and the first origin scaffold, including the corresponding strand and distance. We extend all origins of a group with the shared first extending scaffold. If the extended origins are part of the same valid group, this group is stored as the continuation of the previous group. Groups that are not a continuation for any other group are the ends of the paths and we take the three shared scaffolds (first origin, center, and first extension). Then, we extend these paths until it cannot be continued anymore.

### Testing if two paths can be combined

To test if combining two paths violates the scaffold graph, we take the last scaffolds on the combining side of the two paths. From there, we find all origins matching each path. If no origin matches a path, we take the origins with the longest continuous match starting at the combining end of the path. For every origin on both paths, we take all valid extensions from the origin–extension pairs and search for a full-length match with the other paths. If we found a full-length match, the combination is valid in this direction. If the first scaffold of the other path does not match the extension, the combination is not valid in this direction. If we neither have a full-match nor a complete mismatch, we continue one scaffold into the extension and test the extensions paired with the new origins until we have a decision. If not specified otherwise, we require both directions to be valid, to combine two paths.

If a path has multiple valid combinations, we may search for the best one. For this, we determine all branch points between the alternative paths. Starting from the furthest branch point, we select the scaffold graph entries that reach the other paths and filter out connections, where the remainder (on the other path) does not have a full-length match. If both paths in a combination have alternative connections, we select only the combinations surviving the filtering on both sides.

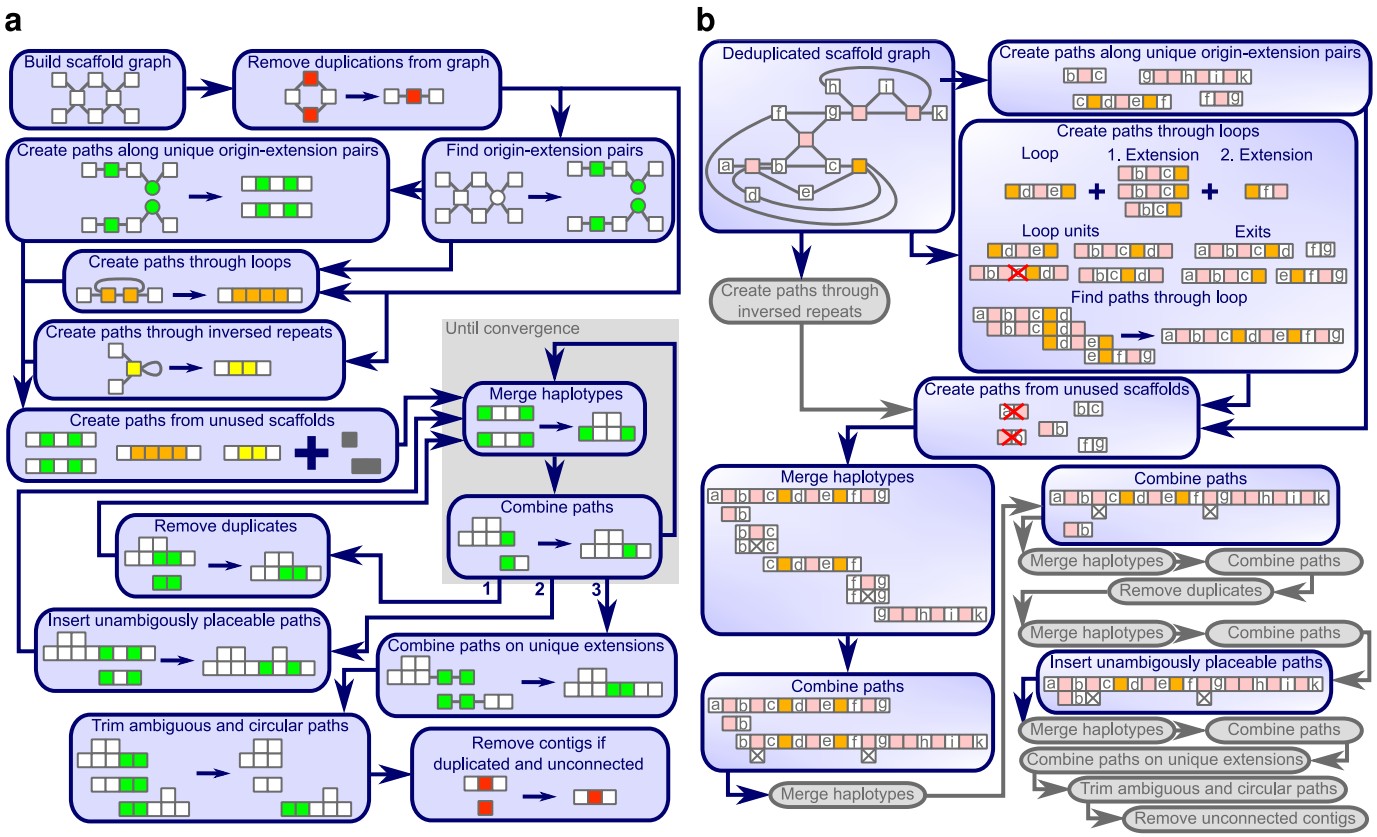

**Figure 4. Overview over the graph traversal.**
**(A)** The functions called during the graph traversal depicted as boxes. **(B)** An example showing the change in each function. Small squares represent scaffolds that are unique (white), duplicated (red), repeated, and traversed (orange) or not traversed (pink) by a single read multiple times, inversely repeated (yellow) or matching between different paths (green). Grey squares could be any of the above. Paths are represented as adjacent squares and scaffolds connected by valid bridges, but not included in the same paths, have lines between them. Crossed out squares in a path are deletions required to align multiple haplotypes. The red crosses over a path, show paths from nonunique origin–extension pairs that are removed, because they are already included in the loop path.

## Creating paths through loops

Loops are identified through scaffolds that appear twice in the same scaffold graph paths. The repeated scaffolds and all scaffolds between them are considered part of the loop. The loop is iteratively extended with scaffolds that are in between any of the already included scaffolds in any scaffold graph paths (example in Fig 4), whereas loops sharing a scaffold are merged.

The exit scaffolds are the first scaffolds in a scaffold graph entry starting in the loop that are not part of the loop. From all exit scaffolds, we take the graph paths entering the loop as exit paths. If the exit paths reach another exit on the other side, the loop is bridged and we take the path from exit to exit. If exactly two unbridged exit scaffolds remain in a loop, we search for a path connecting them.

We start by creating the loop units for each loop by starting from every scaffold inside the loop and greedily extend them along the scaffold graph until the starting scaffold reoccurs. If a path extends to a scaffold outside the loop, it is discarded. For the greedy extension, **gapless** steps ahead in the path one scaffold at a time and takes all scaffold graph paths from that position which are consistent with the already existing paths. If multiple extending paths

are consistent, the path is duplicated and all extensions are followed. Loop units are only accepted if they were found in the greedy extension process in both directions. In the example (Fig 4), one of the four possible loop units is not discovered, because an alternative path had longer connections in the scaffold graph, which caused a greedy decision before the scaffold with a graph entry supporting the not-discovered paths was reached.

Once we have the loop units, we test which exit paths and loop units can be connected according to the path combination test. We allow all combinations, but give a penalty for connections that are not selected as best. To find the path through the loop, we start at one exit and extend one loop unit at a time along valid connections until the other exit is reached. From all the valid paths found, we take the one with the lowest number of penalties and in case of ties the shortest one. These paths together with the bridged paths are added to the initial path set.

## Creating paths through inverted repeats

Inverse repeats are included in the initial set of paths either if they are bridged or if the scaffold graph allows only two exits. In case of unbridged repeats, we additionally filter cases where the exit

scaffold on one side is also present in the scaffold graph path leaving the repeat on the other side, because they also allow the interpretation of two haplotypes on one side and an otherwise unknown exit. After identifying the path through the inverse repeats, we additionally add all other bridged connections between the exits to the initial path set.

### Merging paths

To merge paths that are likely alternative haplotypes, we group all paths that share the scaffolds at both ends. We align all paths in a group pairwise (Fig S15) and split groups if the ends of two paths do not align to each other. This can happen if a path has the end scaffold also included at another position. If a path already has two haplotypes, we treat them as two separate paths within the same group for which we already have the alignment. Within each group, we try to reduce the number of paths to ploidy. First, we remove exactly identical paths; then, we remove paths that only differ by the mean distance in the gaps and if we still have above ploidy alternatives, we remove the paths where scaffolds have only been deleted compared with other paths in the group. The last step handles the case where a scaffold is not well mappable, such that some paths include it and some do not. If we reached ploidy, the paths will be merged; otherwise we keep all non-deleted paths separately. The haplotypes in a path are ordered by the lowest number of supporting reads for any bridge it contains. In case of a tie, we use the next lowest support and so on.

### Combining paths

To combine the paths, we find all overlapping path ends and select the best connection between overlapping paths that can be combined to a valid path. In cases where a path has multiple haplotypes, the best connection is the connection with the most haplotypes considered best connections. If there is only one best connection, two paths are combined. If paths are combined on both sides to another path, we build a group and iteratively combine one path after the other. Before performing the combination, we check again if the paths can be combined, because additional content on the other side could cause the combined paths to violate the scaffold graph.

### Inserting unambiguously placeable paths

The second time convergence is reached in the merge-and-combine loop, we search for the end scaffolds of a given path in all other paths (base) to merge paths if they are not sharing the same ends. If the end scaffolds are present in a base in the correct order and orientation and the base has a free haplotype at that position, we use it as an insertion candidate.

To verify a candidate, we split the base path at the ends of the inserted path, merge the center part with the insertion, and combine all three parts back together. If both merging and combining are valid, the candidate is valid. If this leaves a single valid position for an inserted path, we accept the insertion.

### Combining paths on unique extensions

Sometimes, two combinable paths end up without an overlap. In the finishing procedure, we handle these cases by first continuing all paths with unique extensions. The unique extensions are scaffolds shared between all valid extensions of the origins matching a given side of a path. The extensions are verified at each added scaffold, but are only shortened and never elongated. After we extended all paths, the path combination function from the main loop is run. If two paths were combined, we keep them combined, but remove all additional extensions.

### Trimming duplicated path ends and circular paths

We start at each path end and check how long this end is an exact duplicate of a part of any other path. If the paths of the duplicated parts continue on the same side, it is not an overlap and the duplicated end is separated from the path. If the duplicated ends from both sides of a path overlap, the path is disassembled into its individual scaffolds. Afterwards, we remove completely duplicated paths.

If the end of a path overlaps with the other end of the same path, we have a circular path and remove one of the overlapping ends.

### Aligning reads to the scaffold paths

Once we have the final scaffold paths that describe the (**gapless**) scaffolding of the contigs, we align the long reads to them. The alignment is based on the position of the contigs, to which the long reads map. We require that a read follows a path by mapping to every contig in it until either the read or the path ends. If a gap between two contigs in a path is not covered by any read, we split the path or remove the problematic haplotype.

To remove multi-mapping reads, we count for every contig, how often it is present in a valid alignment of a read. The duplication count of a path alignment is the lowest duplication count of the contigs included in it. Duplication counts of one mean uniquely placed and all alignments with higher counts are removed, except if this would leave a gap in the path uncovered. In this case, the alignments with the lowest duplication count are trimmed to only cover the otherwise uncovered connection.

### Selecting best read to fill into gap

For each gap, we separately look for the best read to fill it, choosing from all reads aligning to this gap in our mapping to the path. The first criterion is the mapping quality on both sides. We first compare the lower of the two and then the higher. Afterwards, we pick the read with the gap length that is the closest to the mean gap length. As a final tie breaker, we use the number of matching nucleotides on each side, again starting with the lower one. In case we have multiple best reads, we pick one arbitrarily.

**Table 2.  Used software versions.**

| Program | Version |
|---------|---------|
| Flye (5) | 2.8.3-b1695 |
| gapless (30) | 0.3 |
| LR_Gapcloser (11) | 156381adec01a5c664edbf5df1d866b5c70e82a1 |
| LRScaf (10) | v1.1.11 Pre-release dfc1617aa9623701878e9b259a5e1c1453faa6fb |
| SAMBA (15) | 6d3ce4e05fe71c8b3ac12ae2f25867f25b80fea4 |
| SH-assembly (23) | 9bb28eb0e5a7492d128c6354be6e56c8ea804900 |
| supernova (22) | 2.1.1 |
| TGS-GapCloser (12) | 1.1.1 8366810b087ed62b4674453479970467927f5191 |
| GNU time | 1.7 |
| ntcard (31) | 1.2.2 |
| Minia (SH-assembly) (32) | 3.2.1 de0334e73cd47487396e406e97b48eccccb12d60 |
| minimap2 (13, 29 *Preprint*) | 2.18-r1015 |
| quast (21) | v5.0.2 |
| racon (17) | v1.4.22 |
| seqtk (16) | 1.3-r106 |
| Snakemake (27) | 3.12.0 |

### Reproducibility of assembly comparison

The benchmark for this article can be run through Snakemake (27) with the scripts available on GitHub (28). All methods were run with default parameters, except specifying the appropriate long-read type. **SAMBA** does not offer a parameter for the read type and **LR_Gapcloser** did not have a setting for PacBio HiFi data, thus we chose the setting for PacBio CLR. **LRScaf** does not include the mapping and we provided it according to its manual with minimap2 alignments, again keeping the default parameters except for specifying the read type. The used minimap2 (13, 29 *Preprint*) version does not yet have a separate setting for PacBio HiFi reads. Thus, we chose -x asm20 according to the minimap2 manual. The reported metrics are from a QUAST-LG (21) comparison to the provided references. For the dolphin assembly, we specified—fragmented and the values for the contigs were obtained with—split-scaffolds. The software versions of all used programs are listed in Table 2.

### Speed and memory benchmarks

We benchmarked the speed and memory usage with GNU time. To report CPU time, we summed user and system times of all individual processes called for an assembly. For maximum memory, we report the highest maximum resident set size of all individual processes. Most processes ran on single-cluster nodes with 384 GB memory and two Intel Xeon Gold 6126 resulting in 48 vCPUs. The dolphin **Flye** and **TGS-GapCloser** assemblies with 86× coverage, the human **Flye** assemblies with 61× and 30× Nanopore coverage, both human **SAMBA** assemblies with 121× Nanopore coverage, both human **gapless** assemblies with 61× Nanopore coverage and the human **TGS-GapCloser** assembly with 30× Nanopore coverage starting from the **supernova** assembly required the high-memory cluster nodes with 3 TB memory and four Intel Xeon CPU E7-4850 v4 resulting in 128 vCPUs. Normally, they were run with only 48 threads, but the dolphin **Flye** assembly with 86× coverage and the human **TGS-GapCloser** assembly with 30× Nanopore coverage starting from the **supernova** assembly needed all 128 cores to finish within the 1 wk limit. The human **Flye** assembly with 121× Nanopore coverage, both human **gapless** assemblies with 121× Nanopore coverage, all four human **TGS-GapCloser** assemblies with 61× and 121× Nanopore coverage, and the human **TGS-GapCloser** assemblies with 15× and 30× Nanopore coverage starting from the **Flye** assembly did not finish within the time limit despite using 128 cores. The dolphin **SAMBA** assemblies with 86× and 5× Nanopore coverage and the human **SAMBA** assembly with 4× HiFi coverage crashed because of an internal error.

## Data Availability

Only public data were used during this study. The *E. coli* PacBio data (ERR1036235) and reference (ERS764956) can be found at: https://www.sanger.ac.uk/resources/downloads/bacteria/nctc/. The Illumina data (SRR3191692) are available from the European Nucleotide Archive. The dolphin data were created by the Vertebrate Genomes Project (19): https://genomeark.github.io/vgp-curated-assembly/Tursiops_truncatus.html. The human data were created by the telomere-to-telomere consortium (assembly version 1.1 and the Nanopore data release 7): https://github.com/marbl/CHM13. **gapless** and all of its code are available under the MIT License at: https://github.com/schmeing/gapless. The Snakemake pipeline and custom scripts used for this publication are available under: https://github.com/schmeing/gapless-benchmark.

# Supplementary Information

# Acknowledgements

The authors thank members of the Robinson Laboratory at the University of Zurich for valuable feedback and the creators of the sequencing data for making them publicly available. This work was supported by the University Research Priority Program (URPP) *Evolution in Action* of the University of Zurich. This work made use of infrastructure provided by S3IT (www.s3it.uzh.ch), the Service and Support for Science IT team at the University of Zurich.

## Author Contributions

S Schmeing: conceptualization, resources, software, formal analysis, methodology, and writing—original draft, review, and editing.
MD Robinson: conceptualization, supervision, funding acquisition, methodology, project administration, and writing—review and editing.

## Conflict of Interest Statement

The authors declare that they have no conflict of interest.

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
