## [Reviewer comments · Life Science Alliance]

Life Science Alliance

Gapless provides combined scaffolding, gap filling and assembly correction with long reads

Stephan Schmeing and Mark Robinson

DOI: <https://doi.org/10.26508/lsa.202201471>

Corresponding author(s): Mark Robinson, University of Zurich

Review Timeline:

Submission Date:	2022-04-03
Editorial Decision:	2022-05-09
Revision Received:	2023-03-14
Editorial Decision:	2023-04-11
Revision Received:	2023-04-17
Accepted:	2023-04-18

Scientific Editor: Novella Guidi

Transaction Report:

May 9, 2022

Re: Life Science Alliance manuscript #LSA-2022-01471-T

Prof. Mark D Robinson
University of Zurich
Institute of Molecular Life Sciences
Winterthurerstrasse 190
IMLS
Zurich, ZH 8057
Switzerland

Dear Dr. Robinson,

Thank you for submitting your manuscript entitled "Gapless provides combined scaffolding, gap filling and assembly correction with long reads" to Life Science Alliance. The manuscript was assessed by expert reviewers, whose comments are appended to this letter. We invite you to submit a revised manuscript addressing the Reviewer comments.

Thank you for this interesting contribution to Life Science Alliance. We are looking forward to receiving your revised manuscript.

Sincerely,

B. MANUSCRIPT ORGANIZATION AND FORMATTING:

Reviewer #1 (Comments to the Authors (Required)):

The authors present a new method for (re-)scaffolding, gap closing and contig correction with long reads. Experiments on existing NGS- or TGS-based draft assemblies show an improvement in contiguity and completeness. And it is more efficient compared to other tools. The algorithm design appears sound and would be useful for the bioinformatics community. The manuscript overall is written and organized well. However, some more analyses and evaluations may be needed. I have the following concerns:

1. The scaffolding algorithm based on the bridges supported by long reads seems convincing. But it is not clear what is the algorithmic improvement compared to previous scaffolding algorithms for paired-end, mate-pair or long reads. Especially for the competitor, SAMBA, which has little effect in the experiments. Plus, it is complicated to compare different tools with various functions all together (e.g., computational performance).
2. It is reasonable to consider the haplotyping and repeat effect in several steps of scaffolding. But the evaluation does not reflect the accuracy of haplotyping or the effect of heterozygosity. Additionally, I would suggest to further output haplotype-specific sequences since the long-range information provided by the long-read alignment has the potential to separate two branches of graph bubbles (for diploid).
3. The gapless further connected Flye-based contigs and showed an improvement in NGA50. What is the reason that gapless builds the inter-contig connections while Flye abandons them? Accuracy evaluations of new connections and inserted sequences are recommended.
4. The algorithm of selecting best read to fill is straightforward. Consensus sequences from the local assembly by mapped long reads may increase the accuracy.
5. The abstract stated that this tool was designed for improving existing assemblies with low-coverage long reads. However, the experiments were conducted also with very high coverages, with which high-quality TGS assembly is possible. Can authors explain the applicable targets? Improving the existing assemblies or de novo extending TGS contigs?
6. It might be helpful to release the tool via conda and Docker as there are several dependencies.

Minor comments:

Here are some examples of erroneous writing as well as the possible correction (sequentially from the start of the paper):

1. Page1 "a read is proposed for gap filling if it has a minimum number of consistent tags on one side of the gap" "a read is proposed for gap filling if it has at least a minimum number of consistent tags on one side of the gap"
2. Page2 "based on a score built from the read's mapping identity and alignment lengths" "based on a score built from the read's mapping identity and alignment length"
3. Page9 "map continuously over the break region are counted as 'vetos'" "map continuously over the break region are counted as 'vetoes'"
4. Page13 "If the exit paths reaches another exit" "If the exit paths reach another exit"
5. Page14 "We align all path in a group pairwise" "We align all paths in a group pairwise"
6. SI Page6 "The datasets are specified in Table ??."

Reviewer #3 (Comments to the Authors (Required)):

In the manuscript "Gapless provides combined scaffolding, gap filling and assembly correction with long reads", Schmeing and Robinson present their computational tool 'gapless'. Gapless combines the tasks of assembly correction, scaffolding and gap filling using long reads and may be useful in cases where genome re-assembly with long reads is not practical or cost-effective.

The manuscript is well-written and all steps of the gapless tools are extensively documented. Schematic illustration of the bridge selection process and refinement of some figures would benefit the readers (detailed below).

Major comments:

While the idea of scaffolding existing (short-read-based) assemblies using long reads is conceptually interesting, the results presented here are not always convincing and may even discourage people from taking this approach.

In most cases, it re-assembling the genome with the long-read data clearly yields far superior results. De novo assembly with Flye for instance, as presented in Fig 2., generally leads to better contiguity and fewer misassemblies than scaffolding the original assembly with the same set of long reads (see Fig2), and in some cases is also less demanding computationally. The argument of not wanting to take the assembly route for want of a list of changes to the initial assembly could be overcome using an assembly-to-assembly mapping and liftover approach. In case computational resources are an issue, it is important to note that much faster assemblers than Flye exist, e.g. the Shasta assembler.

The effect of gapless scaffolding also does not always seem to be readily predictable, and in several cases, adding too much long read data introduces considerable numbers of misassemblies. This is highlighted by the following statement of the authors: "The included assembly correction can, in contrast to other scaffolding tools, remove errors in the initial assembly that are highlighted by the long-reads." As benchmarking shows it can also introduce novel misassemblies, authors should refine and make more explicit their recommendations on when to use gapless. Relatedly, if they expect the original assembler to play a role (see minor comment 3), this should also be taken up in the recommendations.

Minor comments:

- 1) The following statement in the intro requires a reference: "Hybrid assemblies rely either on accurately placing short reads onto long reads for correction or on finding unique paths through the (short-read-generated) De Bruijn graph using the long reads."
- 2) The comparison of computational tools from PBJelly to TSG-GapCloser in the introduction is quite technical and may be better suited for a discussion.
- 3) The poor performance in the authors' benchmarks of the recently published tool SAMBA, which aims to achieve a similar goal, is worrisome. The authors in part attribute this to differences in the software used to generate the initial assemblies. While this may indeed play a role, it would also considerably limit the utility of tools such as SAMBA and gapless. The authors should explore these discrepancies further, e.g. by including MaSuRCA and/or Falcon-based assemblies in their own benchmarks and make recommendations for prospective users.
- 4) It is unclear what the performance of gapless would be on a diploid human genome. The current version of the manuscript only explores human data from the complete hydatidiform mole cell line CHM13, which is effectively haploid. Authors may wish to evaluate gapless performance on HG001 or HG002 (diploid) assemblies and data.
- 5) The details (also their origins) and quality metrics of the initial, unscaffolded assemblies used in the manuscript are lacking, yet represent a key feature to evaluate gapless performance. For instance in Fig 2.
- 6) Figure S1: the panels are overplotted. It is impossible to see the purple triangle in the top left panel. The authors could increase transparency or reduce symbol size. In addition, axes should be fixed to the same range for all panels to avoid inadvertently misleading readers. Vertical lines indicating the lowest attainable and the final selected filtering FDR on the dataset would also be helpful.
- 7) In the evaluation of filtering performance, the word "completeness" is confusing, as it is often used in a different context when evaluating assemblies. Authors may wish to use an alternative or elaborate on their particular usage in the text.
- 8) The effect of the filters is only explored and optimized for a limited range of coverage (21-33x) whereas the data used for the gapless benchmark spans a much wider range (4-121x). Could this explain some of the unexpected decreases in assembly quality as more long read data is used with gapless? If so, this should be either further explored or communicated as recommendations to users.
- 9) In line with the previous comment, as even the most stringent set of bridge filters (which are not the final set being used) can yield ~20% false positive in some cases, it may not be surprising that the amount of misassemblies in the assemblies goes up after gapless scaffolding. Can further filters be included to amend this?
- 10) A schematic overview of the bridge selection process would help clarify this complex step of the algorithm to readers.
- 11) Give the extensive list of software requirements for gapless, the authors may wish to provide a Docker or Singularity image to promote adoption of the tool by the scientific community.

Typos:

In section "Overview of the gapless algorithm"

When this process converges once again, we merge shorter paths into longer ones if we find a unique position and run the process one last time.

In section "Selecting bridges"

For each bin, we count the reads covering the full length and build one cumulative distribution of bins over counts per bin size (Fig. S8, S9).

In section "Creating paths through loops"

If the exit path reaches another exit on the other side, the loop is bridged and we take the path from exit to exit.

We allow all combinations, but give a penalty for connections that are not selected as best.

In section "Merging paths"

To merge paths that are likely alternative haplotypes, we group all paths that share the scaffolds at both ends. We align all paths in a group pairwise (Fig. S11) and split groups if the ends of two paths do not align to each other.

The header of section "Trimming duplicated path ends and circular paths"

Reviewer #1 (Comments to the Authors (Required)):

The authors present a new method for (re-)scaffolding, gap closing and contig correction with long reads. Experiments on existing NGS- or TGS-based draft assemblies show an improvement in contiguity and completeness. And it is more efficient compared to other tools. The algorithm design appears sound and would be useful for the bioinformatics community. The manuscript overall is written and organized well. However, some more analyses and evaluations may be needed. I have the following concerns:

1. The scaffolding algorithm based on the bridges supported by long reads seems convincing. But it is not clear what is the algorithmic improvement compared to previous scaffolding algorithms for paired-end, mate-pair or long reads. Especially for the competitor, SAMBA, which has little effect in the experiments. Plus, it is complicated to compare different tools with various functions all together (e.g., computational performance).

- In this manuscript, we focussed on comparing the final assembly using long reads, which allows long-read assemblers, such as Flye, to be included directly. Including them has not been done in previous publications for scaffolding and gap closing tools, but we consider it important to put the performance into context. The scaffolding and gap closing performance of individual tools can already be accessed in our publication, because we show scaffold and contig performance. To also judge the computational efficiency of individual tools, we added a plot in the supplementary that splits the requirements by tool.

2. It is reasonable to consider the haplotyping and repeat effect in several steps of scaffolding. But the evaluation does not reflect the accuracy of haplotyping or the effect of heterozygosity. Additionally, I would suggest to further output haplotype-specific sequences since the long-range information provided by the long-read alignment has the potential to separate two branches of graph bubbles (for diploid).

- The long-read alignment does have the potential to phase haplotypes, but gapless is the only tool in the comparison that supports the output of haplotypes. Thus, this would require a completely new benchmark that makes comparisons to other tools, and includes other datasets, since only the dolphin dataset is diploid. This benchmark would be a very suitable comparison after an improvement to gapless that also takes SNVs into account for the phasing. However, this lies outside the scope of this publication.

3. The gapless further connected Flye-based contigs and showed an improvement in NGA50. What is the reason that gapless builds the inter-contig connections while Flye abandons them? Accuracy evaluations of new connections and inserted sequences are recommended.

- The Flye contigs are based on the HiFi reads and improved with gapless using Nanopore reads. Thus, no Flye assembly exists that uses the same set of reads and could be directly compared to gapless results.

4. The algorithm of selecting best read to fill is straightforward. Consensus sequences from the local assembly by mapped long reads may increase the accuracy.

- To also improve the contig ends that tend to have an increased error rate in short-read assemblies, a consensus of the full assembly after the insertion of the reads was chosen. An additional consensus for the reads could improve the mapping or reduce the number of times the full consensus needs to be run. To find the optimal consensus strategy for gapless requires further extensive comparisons and lies outside the scope of this publication.

5. The abstract stated that this tool was designed for improving existing assemblies with low-coverage long reads. However, the experiments were conducted also with very high coverages, with which high-quality TGS assembly is possible. Can authors explain the applicable targets? Improving the existing assemblies or de novo extending TGS contigs?

- The tool is designed for improving existing assemblies, but we wanted to provide the full context for readers. How much does additional coverage increase the performance and when should you consider improving the assembly or running a complete *de novo* assembly.

6. It might be helpful to release the tool via conda and Docker as there are several dependencies.

- The software is now available through bioconda.

Minor comments:

Here are some examples of erroneous writing as well as the possible correction (sequentially from the start of the paper):

1. Page1 "a read is proposed for gap filling if it has a minimum number of consistent tags on one side of the gap" "a read is proposed for gap filling if it has at least a minimum number of consistent tags on one side of the gap"

2. Page2 "based on a score built from the read's mapping identity and alignment lengths" "based on a score built from the read's mapping identity and alignment length"

3. Page9 "map continuously over the break region are counted as 'vetos'" "map continuously over the break region are counted as 'vetoes'"

4. Page13 "If the exit paths reaches another exit" "If the exit paths reach another exit"

5. Page14 "We align all path in a group pairwise" "We align all paths in a group pairwise"

6. SI Page6 "The datasets are specified in Table ??."

- Thank you. These errors are fixed now.

Reviewer #3 (Comments to the Authors (Required)):

In the manuscript "Gapless provides combined scaffolding, gap filling and assembly correction with long reads", Schmeing and Robinson present their computational tool 'gapless'. Gapless combines the tasks of assembly correction, scaffolding and gap filling using long reads and may be useful in cases where genome re-assembly with long reads is not practical or cost-effective.

The manuscript is well-written and all steps of the gapless tools are extensively documented. Schematic illustration of the bridge selection process and refinement of some figures would benefit the readers (detailed below).

Major comments:

While the idea of scaffolding existing (short-read-based) assemblies using long reads is conceptually interesting, the results presented here are not always convincing and may even discourage people from taking this approach.

In most cases, it re-assembling the genome with the long-read data clearly yields far superior results. De novo assembly with Flye for instance, as presented in Fig 2., generally leads to better contiguity and fewer misassemblies than scaffolding the original assembly with the same set of long reads (see Fig2), and in some cases is also less demanding computationally. The argument of not wanting to take the assembly route for want of a list of changes to the initial assembly could be overcome using an assembly-to-assembly mapping and liftover approach. In case computational resources are an issue, it is important to note that much faster assemblers than Flye exist, e.g. the Shasta assembler.

- It is true that for the recent, high-quality data from the T2T consortium direct assembly with Flye yields better results than improving a supernova assembly. However, for the PacBio CLR datasets in the case of low coverage (the normal domain of assembly improvement) gapless performs better. Whether to use low coverage nanopore data to improve a HiFi assembly or using direct nanopore assemblies is a trade-off between continuity and correctness. We consider it important to make people aware of the performance comparing direct assembly and assembly improvement. To our knowledge, this has not been done in previous literature (with the exception of SAMBA comparing the results to their own MaSuRCA hybrid assembler). Despite performing worse than Flye for HiFi and Nanopore data, gapless still performs better than the other improvement options. We suspect that the reason for the improved performance of Flye is based on the incorporation of SNV to separate haplotypes and repeat copies, which is most informative if the general error rate is low. This can be added to gapless to resemble better the performance of Flye. Faster direct assemblers do exist, but the independent benchmark cited in the introduction shows that they often perform worse, thus might not be better than assembly improvement.

The effect of gapless scaffolding also does not always seem to be readily predictable, and in several cases, adding too much long read data introduces considerable numbers of

missassemblies. This is highlighted by the following statement of the authors: "The included assembly correction can, in contrast to other scaffolding tools, remove errors in the initial assembly that are highlighted by the long-reads." As benchmarking shows it can also introduce novel misassemblies, authors should refine and make more explicit their recommendations on when to use gapless. Relatedly, if they expect the original assembler to play a role (see minor comment 3), this should also be taken up in the recommendations.

- We explicitly mentioned its usefulness for low-coverage in the abstract and conclusion. We do not want to make an explicit statement that using it on high coverage is discouraged, because it depends on the dataset. We additionally added that <15-fold is what we consider low coverage based on our results to make it more clear.

Minor comments:

1) The following statement in the intro requires a reference: "Hybrid assemblies rely either on accurately placing short reads onto long reads for correction or on finding unique paths through the (short-read-generated) De Bruijn graph using the long reads."

- We added the relevant references.

2) The comparison of computational tools from PBJelly to TSG-GapCloser in the introduction is quite technical and may be better suited for a discussion.

- We think it is beneficial for the reader to know what has been done before to appreciate that the majority of methodology is new.

3) The poor performance in the authors' benchmarks of the recently published tool SAMBA, which aims to achieve a similar goal, is worrisome. The authors in part attribute this to differences in the software used to generate the initial assemblies. While this may indeed play a role, it would also considerably limit the utility of tools such as SAMBA and gapless. The authors should explore these discrepancies further, e.g. by including MaSuRCA and/or Falcon-based assemblies in their own benchmarks and make recommendations for prospective users.

- We reran the assembly from the SAMBA paper and could not reproduce their results. It might be caused by the more recent version we use compared to the original publication. However, this is unlikely to affect the performance of gapless and we did not investigate it further.

4) It is unclear what the performance of gapless would be on a diploid human genome. The current version of the manuscript only explores human data from the complete hydatidiform mole cell line CHM13, which is effectively haploid. Authors may wish to evaluate gapless performance on HG001 or HG002 (diploid) assemblies and data.

- The dolphin assembly included in the benchmark is diploid. More details are already stated in the answer to reviewer #1.

5) The details (also their origins) and quality metrics of the initial, unscaffolded assemblies used in the manuscript are lacking, yet represent a key feature to evaluate gapless performance. For instance in Fig 2.

- The "other" coverage category are the base assemblies improved with scaffolding and gap filling. They are in the same plots (e.g. Fig 2 for comparison). The color states the used assembler for these assemblies.

6) Figure S1: the panels are overplotted. It is impossible to see the purple triangle in the top left panel. The authors could increase transparency or reduce symbol size. In addition, axes should be fixed to the same range for all panels to avoid inadvertently misleading readers. Vertical lines indicating the lowest attainable and the final selected filtering FDR on the dataset would also be helpful.

- Thank you. We improved the figure taking your suggestions into account.

7) In the evaluation of filtering performance, the word "completeness" is confusing, as it is often used in a different context when evaluating assemblies. Authors may wish to use an alternative or elaborate on their particular usage in the text.

- Indeed it could be confusing. To avoid misinterpretation, we renamed completeness to Fraction Possibilities Kept.

8) The effect of the filters is only explored and optimized for a limited range of coverage (21-33x) whereas the data used for the gapless benchmark spans a much wider range (4-121x). Could this explain some of the unexpected decreases in assembly quality as more long read data is used with gapless? If so, this should be either further explored or communicated as recommendations to users.

- We added figures on the filter performance of datasets with very low and very high coverage plots. The conclusions drawn are consistent across all coverage levels. The changes in FDR between the coverage levels do not directly translate into misassemblies detected in the improved assembly.

9) In line with the previous comment, as even the most stringent set of bridge filters (which are not the final set being used) can yield ~20% false positive in some cases, it may not be surprising that the amount of misassemblies in the assemblies goes up after gapless scaffolding. Can further filters be included to amend this?

- We believe that this behaviour is caused by repeats that are not represented in the original assembly as repeats and thus, are not recognized by low mapping quality or the self-mapping of the assembly. To reduce this problem two strategies suggested in the discussion can be used: Including SNV information and mapping to assembly graphs. We added two sentences in the discussion to emphasize the connection between the high FDR and the lower performance compared to Flye.

10) A schematic overview of the bridge selection process would help clarify this complex step of the algorithm to readers.

- We added a supplementary figure with an example of the filtering.

11) Give the extensive list of software requirements for gapless, the authors may wish to provide a Docker or Singularity image to promote adoption of the tool by the scientific community.

- The software is now available through bioconda as suggested by reviewer 1.

Typos:

In section "Overview of the gapless algorithm"

When this process convergeS once again, we merge shorter paths into longer ones if we find a unique position and run the process one last time.

In section "Selecting bridges"

For each bin, we count the reads covering the full length and build one cumulative distributionS of bins over counts per bin size (Fig. S8, S9).

In section "Creating paths through loops"

If the exit pathS reaches another exit on the other side, the loop is bridged and we take the path from exit to exit.

We allow all combinations, but give a penalty for connections that are not selectED as best.

In section "Merging paths"

To merge pathS that are likely alternative haplotypes, we group all paths that share the scaffolds at both ends. We align all pathS in a group pairwise (Fig. S11) and split groups if the ends of two paths do not align to each other.

The header of section "TrimMing duplicated path ends and circular paths"

- Thank you. These errors are fixed now.

April 11, 2023

RE: Life Science Alliance Manuscript #LSA-2022-01471-TR

Prof. Mark D Robinson
University of Zurich
Winterthurerstrasse 190
IMLS
Zurich, ZH 8057
Switzerland

Dear Dr. Robinson,

Thank you for submitting your revised manuscript entitled "Gapless provides combined scaffolding, gap filling and assembly correction with long reads". We would be happy to publish your paper in Life Science Alliance pending final revisions necessary to meet our formatting guidelines.

- please correct the typo in Figure S1 as indicated by Reviewer 3
- please upload your main manuscript text as an editable doc file
- please upload both your main and supplementary as single files and add a separate figure legend section with both your main and supplementary figure legends to your main manuscript text
- please upload your table files as an editable doc or excel file
- please add an abstract, an alternate abstract/summary blurb, and a category for your manuscript to our system
- please add the Twitter handle of your host institute/organization as well as your own or/and one of the authors in our system
- please use the [10 author names, et al.] format in your references (i.e. limit the author names to the first 10)
- please add figure callouts for Figure S3-Figure S6 to your main manuscript text
- please incorporate your conclusion section in your discussion section

A. FINAL FILES:

B. MANUSCRIPT ORGANIZATION AND FORMATTING:

Sincerely,

Reviewer #1 (Comments to the Authors (Required)):

The authors have addressed all my questions and concerns. The paper has been improved after revision.

Reviewer #3 (Comments to the Authors (Required)):

All my comments have been satisfactorily addressed.

There is one small typo in the new Figure S1: "1. Clustering of bridges and filter on absolutE counts"

April 18, 2023

RE: Life Science Alliance Manuscript #LSA-2022-01471-TRR

Prof. Mark D Robinson
University of Zurich
Winterthurerstrasse 190
IMLS
Zurich, ZH 8057
Switzerland

Dear Dr. Robinson,

Thank you for submitting your Research Article entitled "Gapless provides combined scaffolding, gap filling and assembly correction with long reads". It is a pleasure to let you know that your manuscript is now accepted for publication in Life Science Alliance. Congratulations on this interesting work.

DISTRIBUTION OF MATERIALS:

Again, congratulations on a very nice paper. I hope you found the review process to be constructive and are pleased with how the manuscript was handled editorially. We look forward to future exciting submissions from your lab.

Sincerely,
